# Mad1 destabilizes p53 by preventing PML from sequestering MDM2

Jun Wan[1], Samuel Block[2], Christina M. Scribano[1], Rebecca Thiry[1], Karla Esbona[3], Anjon Audhya [2,4] & Beth A. Weaver[1,4,5]

Mitotic arrest deficient 1 (Mad1) plays a well-characterized role in the mitotic checkpoint. However, interphase roles of Mad1 that do not impact mitotic checkpoint function remain largely uncharacterized. Here we show that upregulation of Mad1, which is common in human breast cancer, prevents stress-induced stabilization of the tumor suppressor p53 in multiple cell types. Upregulated Mad1 localizes to ProMyelocytic Leukemia (PML) nuclear bodies in breast cancer and cultured cells. The C-terminus of Mad1 directly interacts with PML, and this interaction is enhanced by sumoylation. PML stabilizes p53 by sequestering MDM2, an E3 ubiquitin ligase that targets p53 for degradation, to the nucleolus. Upregulated Mad1 displaces MDM2 from PML, freeing it to ubiquitinate p53. Upregulation of Mad1 accelerates growth of orthotopic mammary tumors, which show decreased levels of p53 and its downstream effector p21. These results demonstrate an unexpected interphase role for Mad1 in tumor promotion via p53 destabilization.

[1] Department of Cell and Regenerative Biology, University of Wisconsin-Madison, Madison, WI 53705, USA. [2] Department of Biomolecular Chemistry, University of Wisconsin-Madison, Madison, WI 53705, USA. [3] Department of Pathology and Laboratory Medicine, University of Wisconsin-Madison, Madison, WI 53705, USA. [4] Carbone Cancer Center, University of Wisconsin-Madison, Madison, WI 53705, USA. [5] Department of Oncology/McArdle Laboratory for Cancer Research, University of Wisconsin-Madison, Madison, WI 53705, USA. Correspondence and requests for materials should be addressed to B.A.W. (email: baweaver@wisc.edu)

Mad1 was initially discovered in a landmark screen demonstrating that mitosis is regulated by a cell cycle checkpoint, termed the mitotic (or spindle assembly) checkpoint[1]. The mitotic checkpoint ensures accurate chromosome segregation by delaying separation of the replicated sister chromatids until each sister chromatid pair is stably attached to opposite spindle poles through its kinetochores[2–6]. Mad1 plays an evolutionarily conserved role in the mitotic checkpoint by recruiting its binding partner Mad2 to the kinetochores of unattached chromatids[7–9]. At unattached kinetochores, Mad2 is converted into an active mitotic checkpoint inhibitor that delays sister chromatid separation[10–13]. Once the kinetochores of all sister chromatids are stably attached to spindle microtubules, the mitotic checkpoint is satisfied, and Mad1 and Mad2 are no longer recruited. Loss of Mad1 is lethal, and cells with reduced expression of Mad1 missegregate chromosomes to become aneuploid[1,14]. Thus, Mad1 is essential and plays a highly conserved role in ensuring accurate chromosome segregation during mitosis.

Although Mad1 plays a well-characterized role during mitosis, and expression of many mitotic proteins peaks during mitosis, Mad1 expression levels remain constant throughout the cell cycle[2]. During interphase, Mad1 recruits Mad2 to nuclear pores at the nuclear envelope, which permits the production of mitotic checkpoint inhibitors during interphase[3,15–17]. Interphase functions of Mad1 that do not affect mitotic checkpoint signaling have remained largely uncharacterized, although it is known that Mad1 functions independently of Mad2 at the Golgi apparatus to promote secretion of α5 integrin[18,19]. Mad1 is frequently upregulated at both the mRNA and protein level in human breast cancers, where Mad1 upregulation serves as a marker of poor prognosis[2,20,21]. Mad1 upregulation causes a low rate of chromosome missegregation, which is weakly tumor promoting[2,22–24]. However, whether Mad1 upregulation has additional tumor-promoting activities during interphase has remained unclear.

Upregulated Mad1 localizes to nuclear pores and kinetochores, as expected, but also forms punctate structures[2,16]. A fraction of these colocalize with markers of annulate lamellae, storage compartments for excess nuclear pore components, which are predominantly cytoplasmic[2,16,25]. Nuclear Mad1 puncta have remained uncharacterized. Promyelocytic leukemia (PML) nuclear bodies (NBs) represent one prominent source of nuclear puncta. The PML protein, which is fused to retinoic acid receptor alpha (RARα) due to a reciprocal translocation between chromosomes 15 and 17 in >98% of acute PML patients, forms the core of PML NBs[26]. >100 proteins localize to PML NBs, including proteins involved in cell cycle arrest, apoptosis, transcription, and metabolism[27]. Though the proteins that localize to PML NBs are functionally diverse, most of these proteins, including PML itself, are sumoylated[26,27]. Here, we show that upregulated Mad1 localizes to PML NBs.

Protein levels of the p53 tumor suppressor remain low in the absence of cellular stresses due to continuous ubiquitination by MDM2 followed by degradation[28–30]. In response to a variety of cellular stresses including DNA damage, PML sequesters MDM2 in the nucleolus, which physically separates MDM2 from p53 and results in p53 stabilization[31–34]. Here, we demonstrate a previously unexpected interphase role for Mad1 in preventing p53 stabilization. The C-terminal domain (CTD) of Mad1 binds PML directly in a manner facilitated by sumoylation of PML. Upregulated Mad1 localizes to PML NBs, and localization is dependent on the SUMO interacting motif (SIM) within the Mad1 CTD. After DNA damage, upregulated Mad1 displaces MDM2 from PML, replaces MDM2 at nucleoli, and increases the interaction of MDM2 with p53. Mad1-YFP promotes orthotopic mammary tumors in a SIM-dependent manner. These data provide molecular insight into a novel interphase role of Mad1 in destabilizing p53 and promoting tumor initiation and growth.

## Results

**Mad1 accumulates into PML NBs.** Upregulated Mad1 localizes to kinetochores and the nuclear envelope, as expected, but also forms nuclear puncta[2,16]. This fraction of Mad1 does not colocalize with nucleoli (Supplementary Fig. 1a–b), but does show substantial colocalization with Myc and HA tagged SUMO1 and SUMO2 as well as endogenous SUMO1 in MDA-MB-231 breast cancer cells and in HeLa cervical cancer cells (Fig. 1a–c, f, Supplementary Fig. 1b–e). SUMO1 and SUMO2 are highly concentrated in PML NBs[34–36], suggesting that Mad1 puncta localize there. Indeed, Mad1 nuclear puncta substantially colocalize with endogenous as well as HA-tagged PML during both interphase and mitosis (Fig. 1d–f, Supplementary Fig. 1f–g). Under stressful culture conditions, a small portion of endogenous Mad1 also colocalizes with PML (Supplementary Fig. 1h–j).

Arsenic trioxide is therapeutic in acute PML, since it induces proteasomal degradation of PML-RAR, as well as endogenous PML[36–38]. Arsenic treatment was used to confirm that upregulated Mad1 localizes to PML NBs. 12 and 24 h after arsenic treatment, PML protein levels were decreased and PML NBs substantially dispersed (Fig. 1g, h). Substantially fewer Mad1 puncta formed 12 and 24 h after arsenic treatment, although Mad1 protein levels remained constant (Fig. 1g, h, Supplementary Fig. 1k). These data demonstrate that the formation of Mad1 nuclear puncta is dependent upon PML NBs.

Mad1 recruits Mad2 to unattached kinetochores and nuclear pores[3,16,39], and Mad1 interacts with Mad2 throughout the cell cycle[3,40]. Mad2 is expressed in molar excess of Mad1, and the only known pool of Mad1 that does not recruit Mad2 is at the Golgi apparatus[41]. To determine whether Mad1 recruits Mad2 to PML NBs, Mad1-3xFLAG and Mad2-GFP were expressed singly or in combination. Mad1-3xFLAG recruited Mad2-GFP to nuclear puncta, which Mad2-GFP did not form when expressed alone (Supplementary Fig. 1l). Thus, Mad1 recruits Mad2 to PML NBs, as it does to unattached kinetochores and nuclear pores.

We have previously shown that Mad1 protein levels are commonly increased in breast cancer patients, and that upregulated Mad1 forms punctate structures in breast cancer[2]. To determine if Mad1 puncta represent PML NBs, sections from nine primary breast cancers were costained with antibodies to Mad1 and SUMO1. Indeed Mad1 puncta frequently colocalized with SUMO1 (Fig. 1i, j). Together, these results demonstrate that upregulated Mad1 localizes to PML NBs in cultured cells and in human cancer.

**Mad1 interacts with PML through its C terminal domain (CTD).** As an initial test of whether Mad1 localization to PML NBs was due to an interaction with PML or with another component of PML NBs, Mad1-3xFLAG was immunoprecipitated from HEK293T cells. HA-PML was co-immunoprecipitated with Mad1-3xFLAG, supporting an interaction between these proteins (Supplementary Fig. 1m). To determine the region of Mad1 responsible for the interaction with PML, 10 deletion mutants of Mad1 were generated (Fig. 2a, Supplementary Fig. 2a). Immunoprecipitation experiments revealed that all constructs containing the CTD of Mad1 (aa 597–718) co-precipitated PML, while fragments lacking the Mad1 CTD did not (Fig. 2b). A reciprocal immunoprecipitation experiment showed that immunoprecipitation of HA-PML co-precipitated all Mad1 fragments containing the CTD, but no fragments lacking the Mad1 CTD (Fig. 2c). Thus, Mad1 interacts with PML in cell extracts through its CTD.

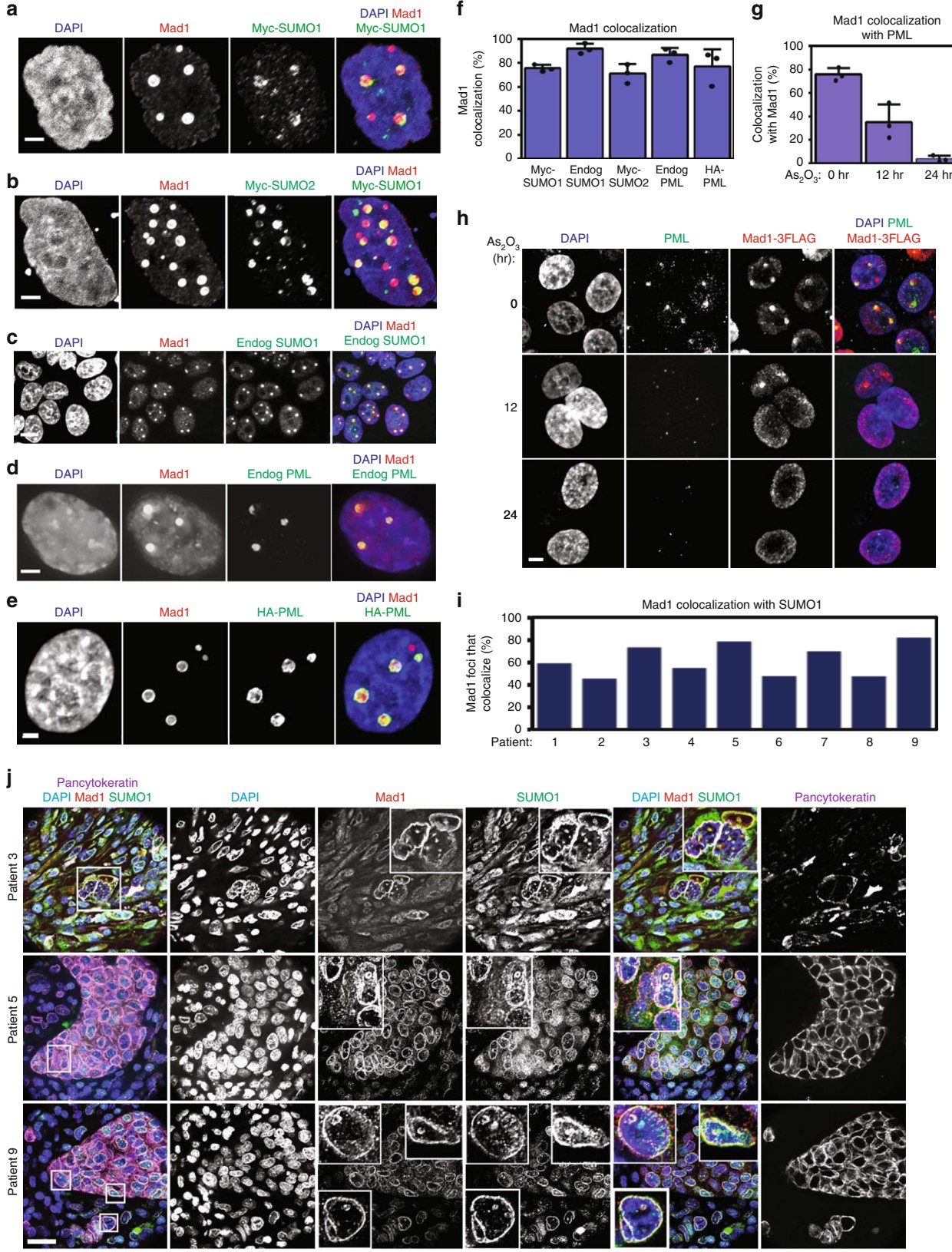

We then tested the subcellular localization of the Mad1 fragments (Fig. 2d–f, Supplementary Fig. 2b–d). While full length Mad1 localized to PML NBs, a fragment of Mad1 lacking the CTD (aa 1–596) was excluded from PML NBs (Fig. 2d, e), consistent with the requirement of the Mad1 CTD for the interaction of Mad1 with PML. A fragment of Mad1 that contains the CTD and co-immunoprecipitates with PML, aa 180–718, is cytoplasmic and therefore unable to localize to PML NBs, presumably because it lacks the endogenous nuclear localization sequence (NLS; Fig. 2e, Supplementary Fig. 2b). The addition of the NLS from SV40 restored PML localization of aa 180–718 (Fig. 2e), but not of fragments lacking the CTD containing aa

**Fig. 1** Upregulated Mad1 accumulates into PML nuclear bodies (NBs) in cells and tumors. **a–c** Mad1-3xFLAG colocalizes with SUMO1 and SUMO2, which are markers of PML nuclear bodies. Mad1-3xFLAG colocalizes with Myc-SUMO1 (**a**) and Myc-SUMO2 (**b**) in MDA-MB-231 cells. Scale bars, 2.5 μm. **c** Mad1-3xFLAG colocalizes with endogenous SUMO1 in HeLa cells. Scale bar, 20 μm. **d** Mad1-3xFLAG colocalizes with endogenous PML in HeLa cells. Scale bar, 2.5 μm. **e** Mad1-3xFLAG colocalizes with HA-PML in HeLa cells. Scale bar, 2.5 μm. **f** Quantitation (±SD) of the percentage of cells with Mad1-3xFLAG puncta that colocalize with Myc-SUMO1, Myc-SUMO2, endogenous SUMO1, endogenous PML, and HA-PML, as shown in panels (**a–e**). $n = 200$ cells from each of three independent experiments. **g** HeLa cells expressing Mad1-3xFLAG in a tet-inducible manner were treated with tet for 24 h followed by treatment with 1 μM $As_2O_3$ to disintegrate PML nuclear bodies for the indicated time. Quantitation (±SD) of the percentage of PML foci that colocalize with Mad1 in the presence or absence of $As_2O_3$. $n = 200$ cells from each of three independent experiments. **h** Immunofluorescence showing loss of PML NBs and Mad1 puncta in response to arsenic. Scale bar, 5 μm. **i, j** Mad1 localizes to PML NBs in human breast cancer tumor tissue sections. **i** Quantitation of the percentage of Mad1 puncta in breast tumors that colocalize with SUMO1, a marker of PML NBs. $n = 100$ cells from each of 9 independent samples. **j** Representative images of Mad1 puncta colocalizing with SUMO1 in breast tumors from subjects 3, 5, and 9. Pan-cytokeratin is used as a marker of epithelial (tumor) cells. Insets show a magnified view of cells boxed in the first column. Scale bar, 100 μm

180–480 or aa 180–596 (Supplementary Fig. 2d). The Mad1 CTD (aa 597–718) domain is not sufficient to support localization to PML NBs, at least in part due to lack of an NLS (Fig. 2f). Unexpectedly, the addition of the SV40 NLS drove localization of the CTD to the nuclear envelope, but not to PML NBs; fusion with the endogenous nuclear pore targeting domain (NPD; aa 1–274[3]) was necessary to localize the Mad1 CTD to PML NBs (Fig. 2f).

To define the domain of PML that interacts with Mad1, four truncations of PML were generated (Fig. 2g). Immunoprecipitation of HA-tagged full length and aa 1–229 of PML co-precipitated Mad1-3xFLAG (Fig. 2h). Reciprocal immunoprecipitation of Mad1 confirmed that Mad1-3xFLAG interacts with aa 1–229 of PML (Fig. 2i). Together, these data demonstrate that the C-terminus of Mad1 interacts with the N-terminus of PML.

**A conserved Mad1 SIM is critical for localization with PML.** The N-terminal domain of PML contains two sumoylation sites (Fig. 2g)[42,43]. SUMO chains at these sites interact with the SIM of other proteins in PML NBs[42]. To determine whether the interaction of Mad1 with PML involved sumoylation, we performed bioinformatic analysis of the structure of Mad1. Using the GPS–SUMO algorithm[44,45] we identified a putative SIM in the CTD of Mad1 (aa 689–699; Fig. 3a). The putative SIM in the Mad1 CTD is conserved across multiple species (Fig. 3a) and similar to the bona fide SIMs in PML and the PML interaction partner DAXX (Fig. 3b). The SIM consists of a hydrophobic core separated from an acidic stretch by a short spacer region[46,47]. Structural analysis and visualization of the Mad1 CTD using SWISS-MODEL Workspace (4dzo.1) and PyMOL software[48], revealed a hydrophobic surface localized within the putative SIM at aa 689–692 (Supplementary Fig. 3a). This result is consistent with the presence of the predicted SIM in the CTD of Mad1.

To determine the importance of the SIM in the CTD of Mad1 for localization to PML NBs, the leucine (aa 689) and isoleucine (aa 690) were mutated into lysine residues to abolish the hydrophobic core (Fig. 3b). Mutation of LIEV to KKEV (hereafter SIM mutant) substantially altered Mad1 localization, dispersing it from PML NBs to numerous smaller puncta (Fig. 3c), suggesting that the SIM is essential for Mad1 to localize to PML NBs. Consistent with this, mutation of the SIM in the CTD of Mad1 substantially decreased the interaction of Mad1 with PML in reciprocal immunoprecipitation experiments (Fig. 3d). These experiments demonstrate that mutation of two residues within the CTD of Mad1 largely disrupts its interaction with PML.

**The CTD of Mad1 interacts directly with PML.** To determine whether the interaction between Mad1 and PML is direct, MBP tagged full length Mad1 or the Mad1 CTD were co-expressed with recombinant PML in bacteria (Fig. 3e, Supplementary Fig. 3b). MBP alone, MBP-Mad1 (full length), or MBP-Mad1

(CTD) was isolated from cell lysates with amylose resin and binding to PML was evaluated by SDS-PAGE and western blot. This analysis revealed that full length Mad1 and the Mad1 CTD both interact directly with PML (Fig. 3f). To determine the importance of sumoylation in the interaction between Mad1 and PML, three copies of SUMO-2 were linked to the N-terminus of PML (Fig. 3e, Supplementary Fig. 3b). The addition of 3xSUMO-2 enhanced binding of both full length Mad1 and the CTD of Mad1 to PML (Fig. 3f). These data demonstrate that Mad1 binds directly to PML and that this interaction is enhanced by sumoylation.

**Mad1 upregulation prevents stabilization of p53.** PML plays an important role in stabilizing the p53 tumor suppressor in response to cellular stress[31,34,43]. Considering that Mad1 is frequently upregulated in human breast cancers[2], and that upregulated Mad1 localizes to PML NBs, we tested whether Mad1 upregulation affects p53 stabilization. Tetracycline (tet)-inducible expression of Mad1 prevented the increase in p53 protein levels observed in response to DNA damage caused by the topoisomerase II inhibitor doxorubicin in control cells in multiple cancer cell types (MDA-MB-231, HeLa, DLD1; Fig. 4a, b, Supplementary Fig. 4a). These cell lines all have an impaired p53 pathway, due to an R280K mutation in p53, expression of HPV E6, or S241F mutation, respectively[49–51]. Upregulation of Mad1 also prevented stabilization of p53 in response to DNA damage in nontransformed MCF10A breast cells, which express wild type p53[51,52] (Fig. 4c).

p21 is a critical mediator of p53-dependent cell cycle arrest in response to DNA damage[53–55], and induction of p21 upon DNA damage is regulated by PML[31,34,56]. As an initial test of whether upregulation of Mad1 affects p53 downstream signaling, p21 expression levels were assessed. Mad1 upregulation prevented p21 accumulation in nontransformed MCF10A (Fig. 4c) as well as cancerous HeLa (Fig. 4d, Supplementary Fig. 4b) cells.

To determine if the role of Mad1 in regulating the p53 signaling pathway is dependent on its interaction with PML, Mad1 lacking the CTD or the SIM was tested. Expression of Mad1 lacking the CTD restored stabilization of p53 and p21 in response to DNA damage (Fig. 4b, Supplementary Fig. 4b). Similarly, both p53 and p21 accumulated in cells expressing the Mad1 SIM mutant, similar to control cells (Fig. 4d). These data demonstrate that the interaction with PML is necessary for upregulated Mad1 to prevent stabilization of p53.

To further evaluate the effect of Mad1 upregulation on p53 signaling, cell death in response to DNA damage was assessed. We generated cell lines stably expressing 3xFLAG-tagged full length wild type Mad1, Mad1 lacking the CTD, or SIM mutant Mad1 in a tet-inducible manner. DNA damage induced cell death in control cells (Fig. 4e, Supplementary Fig. 4c). Expression of full length Mad1 substantially reduced cell death,

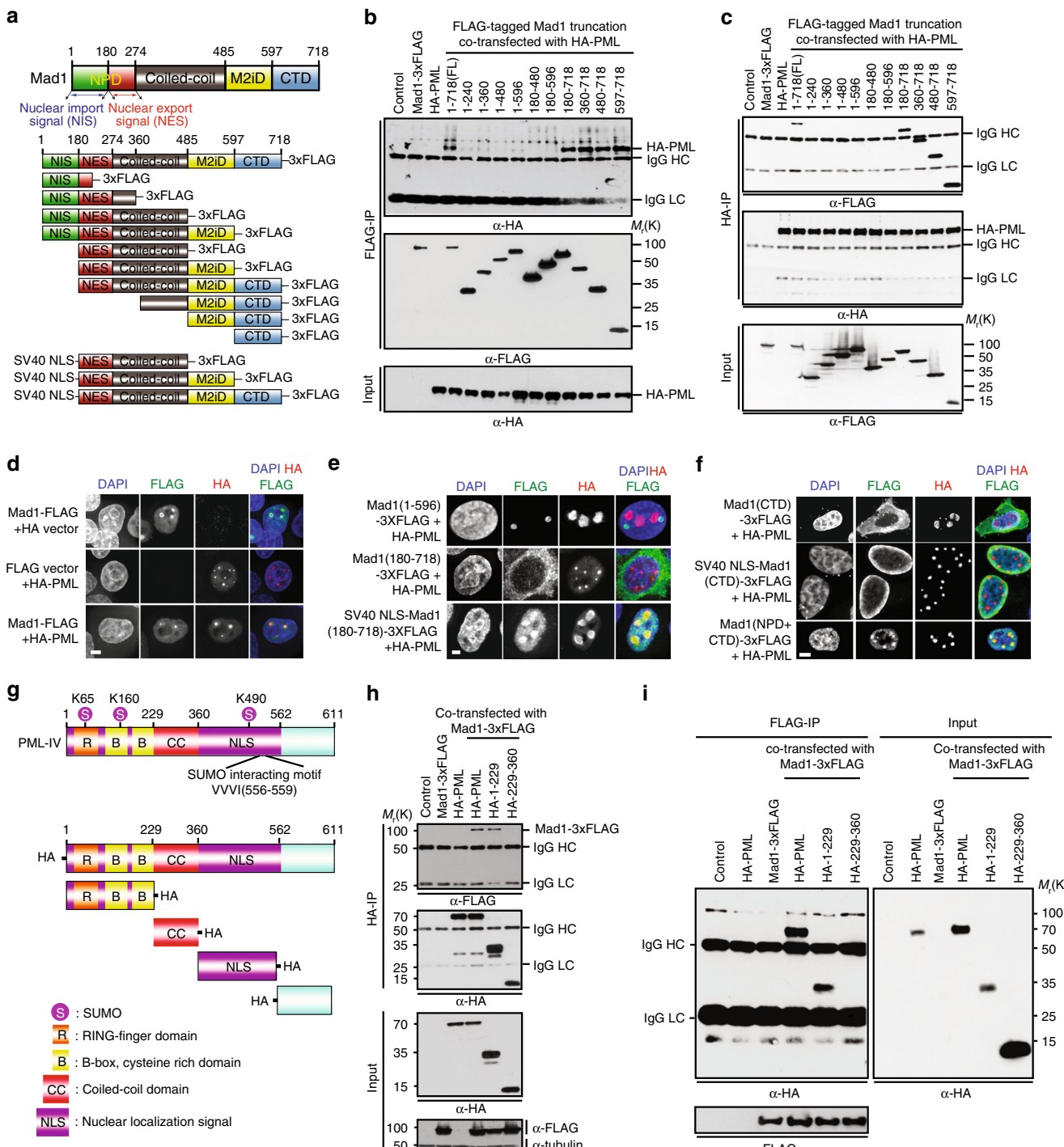

**Fig. 2** The C-terminal domain (CTD) of Mad1 interacts with the N-terminus of PML. **a** Schematic of Mad1 and the Mad1 fragments tested. NIS nuclear import signal, NPD nuclear pore targeting domain[3], M2iD Mad2-interacting domain[11,70], CTD C terminal domain[70]. **b, c** The CTD of Mad1 is necessary for Mad1 to interact with PML. 1–718(FL) indicates full length Mad1. **b** Only Mad1-3xFLAG constructs containing the CTD co-immunoprecipitate HA-PML from 293T cells. Blot is representative of 3 independent experiments. **c** HA-PML co-immunoprecipitates all FLAG-tagged Mad1 fragments that contain the CTD, but not fragments lacking the CTD. Blot is representative of 3 independent experiments. **d–f** The CTD of Mad1 is necessary for its localization to PML NBs. HeLa cells were co-transfected with constructs for full length wild type PML-HA and full length wild type Mad1-3xFLAG or the indicated FLAG-tagged Mad1 deletion mutants and analyzed by immunofluorescence microscopy with anti-HA and anti-FLAG antibodies. Scale bars, 5 μm. **d** Full length Mad1-3xFLAG localizes to PML NBs. **e** The CTD of Mad1 is necessary for localization to PML NBs. **f** The CTD also requires the Mad1 nuclear pore targeting domain (NIS + NES; aa 1–274)[3] for localization to PML NBs. **g–i** The N-terminus of PML interacts with Mad1. **g** Schematic of PML-IV and the fragments used in this study. R RING-finger domains, B B-boxes, CC α-helical coiled-coil domain, NLS nuclear localization signal, S SUMOylation site[43]. Only the N-terminal and CC domains were efficiently expressed. **h** Full length PML and an N-terminal fragment but not an internal fragment co-immunoprecipitate Mad1-3x-FLAG. Blot is representative of 3 independent experiments. **i** Mad1-3xFLAG co-immunoprecipitates full length and the N-terminus of PML, but not an internal fragment of PML. Blot is representative of 3 independent experiments

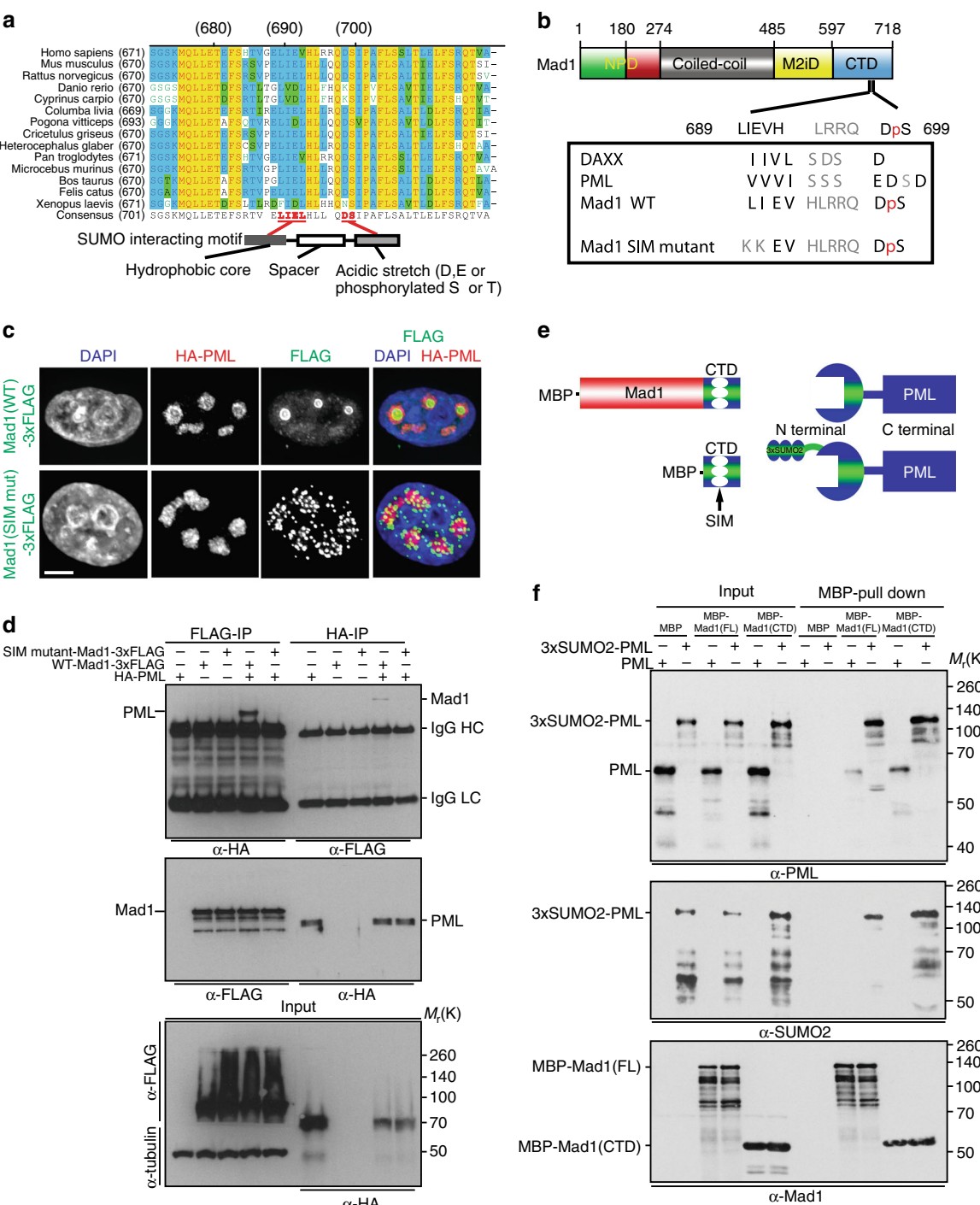

**Fig. 3** A conserved SIM within the Mad1 CTD is necessary for Mad1 localization to PML NBs. **a** Alignment of a portion of the Mad1 CTD protein sequence showing a conserved putative SIM. Residues identical in all species are shown in yellow. Residues conserved in a majority of species are shown in blue. Functional components of SIMs are shown in the schematic underneath. Putative Mad1 SIM residues conserved between species are shown in bolded red font. **b** Sequence alignment of the putative SIM within the Mad1 CTD and known SIMs in PML and DAXX. Red lowercase p indicates the serine is phosphorylated. **c** Mutation of the SIM disperses Mad1 from PML NBs. HeLa cells co-transfected with FLAG-tagged wild type (WT) or SIM mutant Mad1 and HA-tagged PML were analyzed 36 h after transfection using Mad1 and HA antibodies. The leucine and isoleucine residues at positions 689 and 690 were mutated to lysines to generate SIM mutant Mad1. Scale bar, 2.5 μm. **d** Reciprocal immunoprecipitations showing the interaction between Mad1 and PML is dependent on the Mad1 SIM. 293T cells were co-transfected with HA-tagged PML and FLAG-tagged WT or SIM mutant Mad1. 48 h later, cells were lysed and FLAG or HA antibodies used for immunoprecipitation. A single film for both experiments is shown. Blot is representative of 3 independent experiments. **e** Schematic of proteins and protein fragments used in (**f**). Sumoylation of PML was simulated by fusing a 3xSUMO2 chain to the N terminus of PML. **f** MBP pull-down experiments showing both full length Mad1 and the CTD of Mad1 interact with PML directly and that this interaction is enhanced by sumoylation of PML

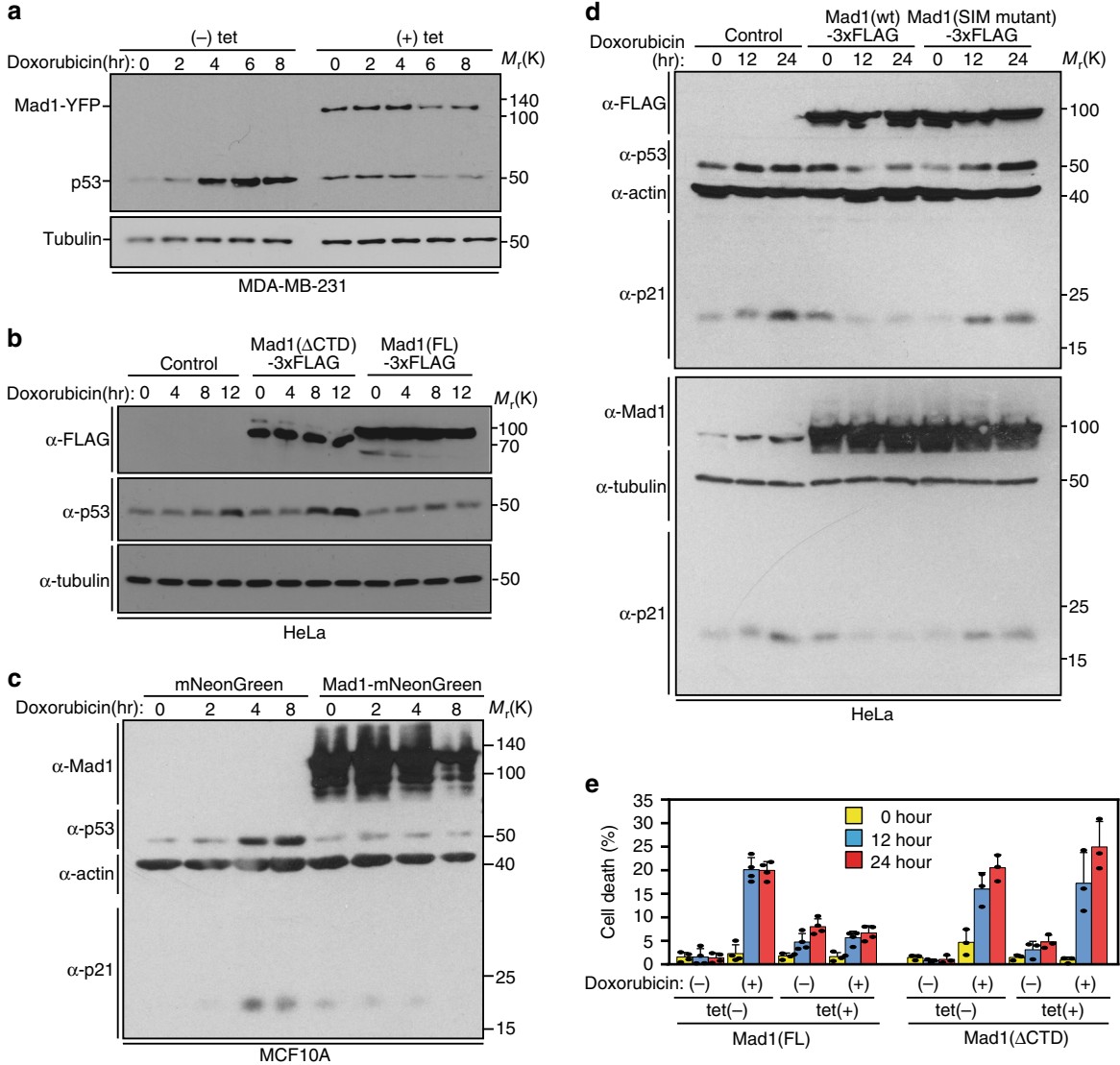

**Fig. 4** Mad1 upregulation destabilizes p53 and impairs cell death in response to DNA damage. **a** MDA-MB-231 cells stably expressing Mad1-YFP in response to tet were treated ± tet for 24 h and then with the topoisomerase II inhibitor doxorubicin (2 μg/mL) to induce DNA damage for the indicated times. Blot is representative of 3 independent experiments. **b** The CTD of Mad1 is essential for its role in preventing stabilization of p53. HeLa cells stably expressing full length (FL) Mad1-3xFLAG or Mad1 lacking the CTD (aa 1–596) in a tet-inducible manner were treated with tet for 24 h and then with doxorubicin (2 μg/mL) for the indicated times. Blot is representative of 3 independent experiments. **c** MCF10A cells were infected with adenoviruses expressing mNeonGreen or Mad1-mNeonGreen for 1 h and then treated with 1 μg/mL doxorubicin to induce DNA damage for the indicated times. Expression of Mad1-mNeonGreen prevents stabilization of p53 and its downstream effector p21. **d** The SIM in the CTD of Mad1 is necessary for Mad1 to prevent stabilization of p53 and p21. HeLa cells stably expressing tet-inducible wild type Mad1-3xFLAG or Mad1-SIM mutant-3xFLAG were treated with tet for 24 h and then with 2 μg/mL doxorubicin for the indicated number of hours. Wild type Mad1 but not SIM mutant Mad1 prevents stabilization of p53 and p21. Blot is representative of 3 independent experiments. **e** Quantitation of cell death (±SD) in HeLa cells stably expressing tet-inducible 3xFLAG tagged full length or ΔCTD Mad1 ± 24 h tet, then ±2 μg/mL doxorubicin for 0, 12, and 24 h. Full length Mad1, but not Mad1 lacking the CTD, prevented cell death due to DNA damage. $n > 250$ cells from each of 3 independent experiments

while expression of Mad1 lacking the CTD or SIM did not (Fig. 4e, Supplementary Fig. 4c). Thus, upregulation of Mad1 impairs p53 stabilization and downstream signaling, and the interaction of Mad1 with PML is necessary for these effects.

Importantly, upregulation of endogenous Mad1 also prevents p53 stabilization in a breast cancer cell line that expresses high levels of Mad1 protein without experimental manipulation (SKBR3; Supplementary Fig. 5a). Consistent with its localization in primary breast tumors and after induced expression, Mad1 localizes to PML NBs in SKBR3 cells (Supplementary Fig. 5b). Partial depletion of Mad1 with four different shRNA sequences increases the level of p53 in response to DNA damage

(Supplementary Fig. 5c). Thus, upregulation of endogenous or exogenous Mad1 results in reduced protein levels of p53.

**Mad1 displaces MDM2 from nucleoli after DNA damage**. p53 levels are normally low due to ubiquitination by MDM2 followed by degradation. In response to DNA damage, PML sequesters MDM2 in the nucleolus, which physically separates MDM2 from p53 and allows p53 protein levels to accumulate[31]. To determine whether Mad1 destabilizes p53 by disrupting MDM2 translocation into nucleoli, we examined MDM2 localization in cells that stably express Mad1 in response to tet. After DNA damage, a

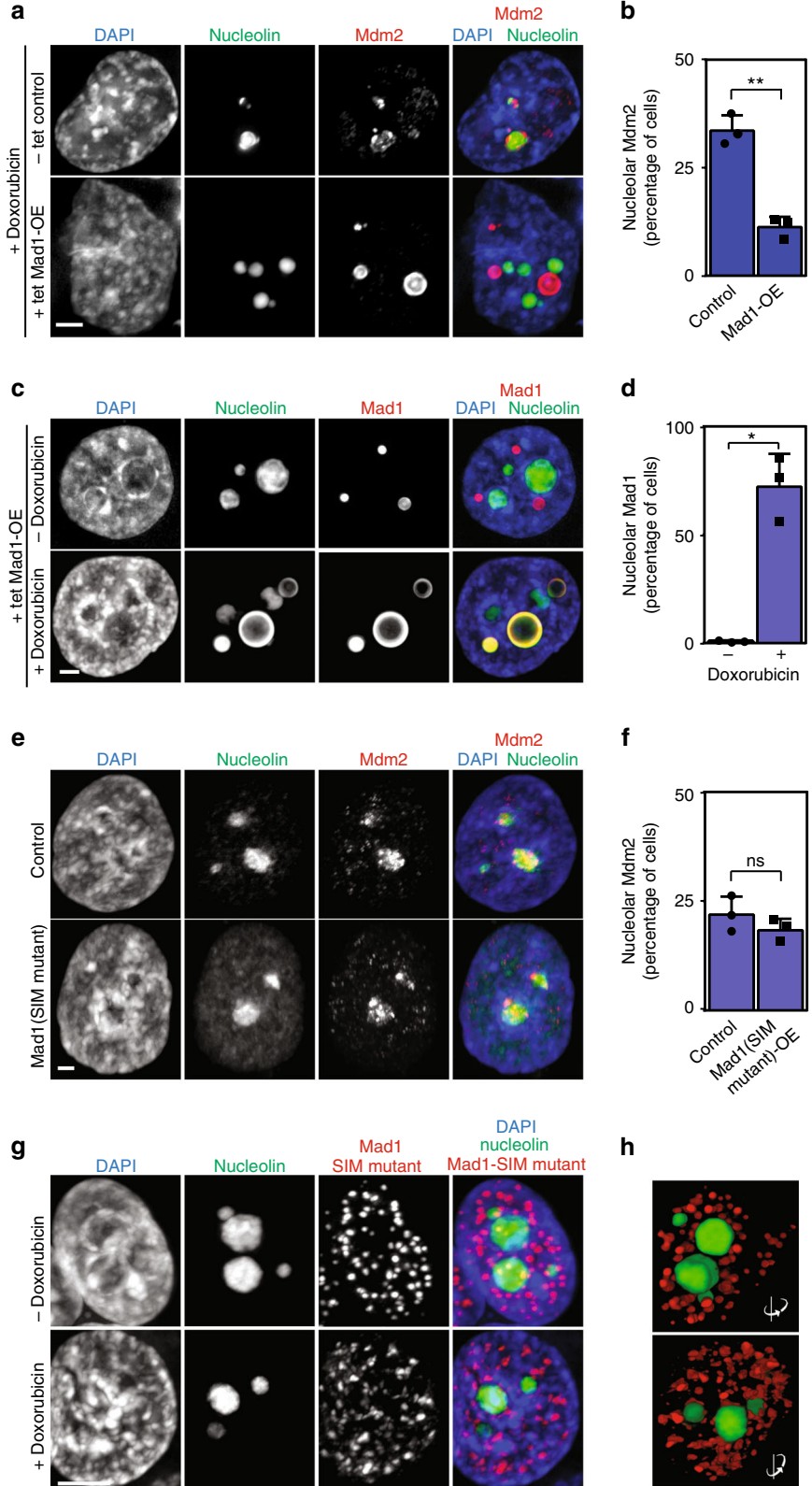

substantial portion of MDM2 colocalizes with nucleolin (Fig. 5a, b). In contrast, in cells expressing full length wild type Mad1 in response to tet, MDM2 localization to nucleoli is substantially reduced (Fig. 5a, b). Conversely, while Mad1 expressed in response to tet does not colocalize with nucleolin in the absence of DNA damage, in the presence of doxorubicin a majority of

cells contain nucleolar Mad1 (Fig. 5c, d). Because the SIM of Mad1 contributes to its interaction with PML (Fig. 3c–f), we predicted that SIM mutant Mad1 would be unable to displace MDM2 from nucleoli. Indeed, expression of SIM mutant Mad1 did not affect localization of MDM2 to nucleoli in the presence of DNA damage (Fig. 5e, f), nor did SIM mutant Mad1 localize to

**Fig. 5** Mad1 destabilizes p53 by preventing sequestration of MDM2 into nucleoli. **a–d** In response to DNA damage, upregulated Mad1 replaces MDM2 in nucleoli. **a, b** MDM2 relocalizes to the nucleolus after DNA damage caused by doxorubicin in control cells (-tet) but not in cells expressing Mad1 in response to tet. HeLa cells stably expressing tet-inducible full length wild type Mad1 and GFP-nucleolin-P2A-3xFLAG-MDM2 were cultured for 24 h in the presence of tet and an additional 8 h with 2 μg/mL doxorubicin. **a** Immunofluorescence showing colocalization of MDM2 with GFP-nucleolin after DNA damage. Scale bar, 2.5 μm. **b** Quantification of the percentage of cells (±SD) exhibiting nucleolar MDM2. $n > 200$ cells from each of three independent experiments. **c, d** Upregulated Mad1 localizes to nucleoli in response to DNA damage. HeLa cells stably expressing GFP-nucleolin and tet-inducible wild type Mad1 were cultured with tet for 24 h and doxorubicin for another 8 h. **c** Immunofluorescence showing colocalization between Mad1 and GFP-nucleolin in response to DNA damage. Scale bar, 2.5 μm. **d** Quantitation of the percentage of cells (±SD) with nucleolar Mad1 staining. $n \geq 200$ cells from each of three independent experiments. **e, f** HeLa cells stably expressing tet-inducible Mad1(SIM mutant)-3xFLAG were incubated with adenovirus expressing nucleolin-mNeonGreen-P2A-mScarlet-MDM2 for 1 h and then cultured with tet for 24 h and doxorubicin for another 8 h. **e** Immunofluorescence showing MDM2 localization to nucleoli after doxorubicin was unaffected by expression of SIM mutant Mad1. **f** Quantification of the percentage of cells (±SD) exhibiting nucleolar MDM2 in (**c**). $n > 200$ cells from each of three independent experiments. **g, h** HeLa cells stably expressing tet-inducible SIM mutant Mad1 were incubated with adenovirus expressing mNeonGreen-nucleolin for 1 h and then cultured for 24 h in the presence of tet and an additional 8 h ± 2 μg/mL doxorubicin. **g** SIM mutant Mad1 does not colocalize with nucleoli. Scale bar, 5 μm. **h** Rotated views of (**g**), showing the lack of colocalization between SIM mutant Mad1 and GFP-nucleolin. *$p < 0.05$. **$p < 0.001$. ns non-significant using $t$ test. For specific $p$ values, see Source Data file

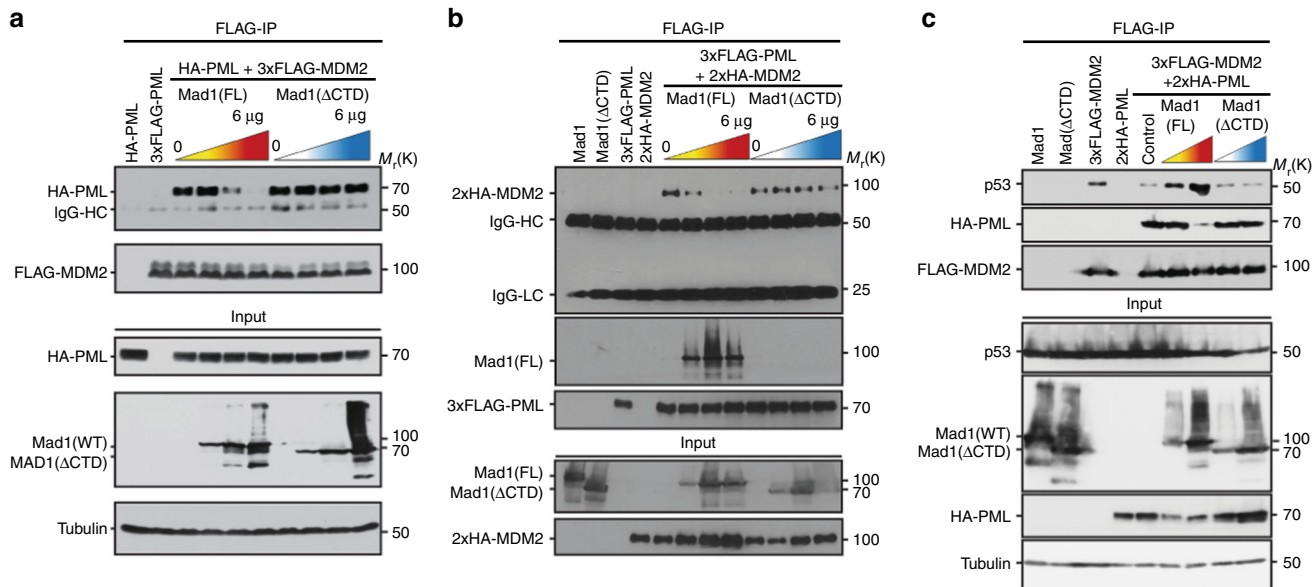

**Fig. 6** Upregulation of Mad1 prevents the interaction between MDM2 and PML, freeing MDM2 to bind p53. **a** Increased expression of full length (FL) Mad1 but not Mad1 lacking the CTD (aa 1–596; Mad1ΔCTD) impairs co-immunoprecipitation of PML with MDM2. 293T cells were co-transfected with HA-PML, 3xFLAG-MDM2, and increasing amounts of FL Mad1 or Mad1ΔCTD. **b** Reciprocal experiment showing that increased expression of FL Mad1 but not Mad1ΔCTD impairs co-precipitation of MDM2 with PML. 293T cells were co-transfected with 3xFLAG-PML, 2xHA-MDM2, and increasing amounts of FL Mad1 or Mad1ΔCTD. **c** Upregulation of Mad1 inhibits the interaction of MDM2 with PML, freeing MDM2 to interact with p53. 293T cells were co-transfected with 2xHA-PML, 3xFLAG-MDM2, and increasing amounts of FL Mad1 or Mad1ΔCTD. **a–c** 48 h after transfection, cell extracts were prepared and immunoprecipitated using beads coupled to anti-FLAG antibodies. Blots are representative of 3 independent experiments

nuclei after doxorubicin treatment (Fig. 5g, h). These data demonstrate that upregulated Mad1 displaces MDM2 from nucleoli in response to DNA damage in a SIM-dependent fashion.

**Mad1 regulates the interaction of MDM2 with PML and p53.** In response to DNA damage, PML escorts MDM2 to nucleoli through a direct interaction, which separates MDM2 from p53 and permits p53 protein levels to accumulate[31,33]. To determine whether Mad1 upregulation prevents p53 stabilization by displacing MDM2 from PML, MDM2 was immunoprecipitated in the presence of increasing amounts of full length Mad1 or Mad1 lacking the CTD. Expression of full length Mad1 decreased the co-immunoprecipitation of PML with MDM2 in a dose-dependent fashion (Fig. 6a). Importantly, the expression of Mad1 lacking the CTD to a similar level as full length Mad1 did not affect the ability of MDM2 to co-immunoprecipitate PML

(Fig. 6a). Reciprocal experiments, in which PML was immunoprecipitated, showed that co-immunoprecipitation of MDM2 with PML was substantially impaired by full length Mad1 but not by Mad1 lacking the CTD (Fig. 6b). While full length Mad1 was co-immunoprecipitated with PML, Mad1 lacking the CTD was not, despite similar expression levels (Fig. 6b). These results demonstrate that Mad1 binding displaces MDM2 from PML.

MDM2 immunoprecipitations were also used to determine whether the MDM2 displaced by Mad1 can interact with p53. Expression of full length Mad1, but not Mad1 lacking the CTD, impaired co-precipitation of PML with MDM2 (Fig. 6c), consistent with the previous result (Fig. 6a). Importantly, expression of full length Mad1 increased the amount of p53 co-precipitated with MDM2, while expression of Mad1 lacking the CTD did not (Fig. 6c). These results demonstrate that Mad1 binding to PML displaces MDM2 from PML, and that the displaced MDM2 binds p53.

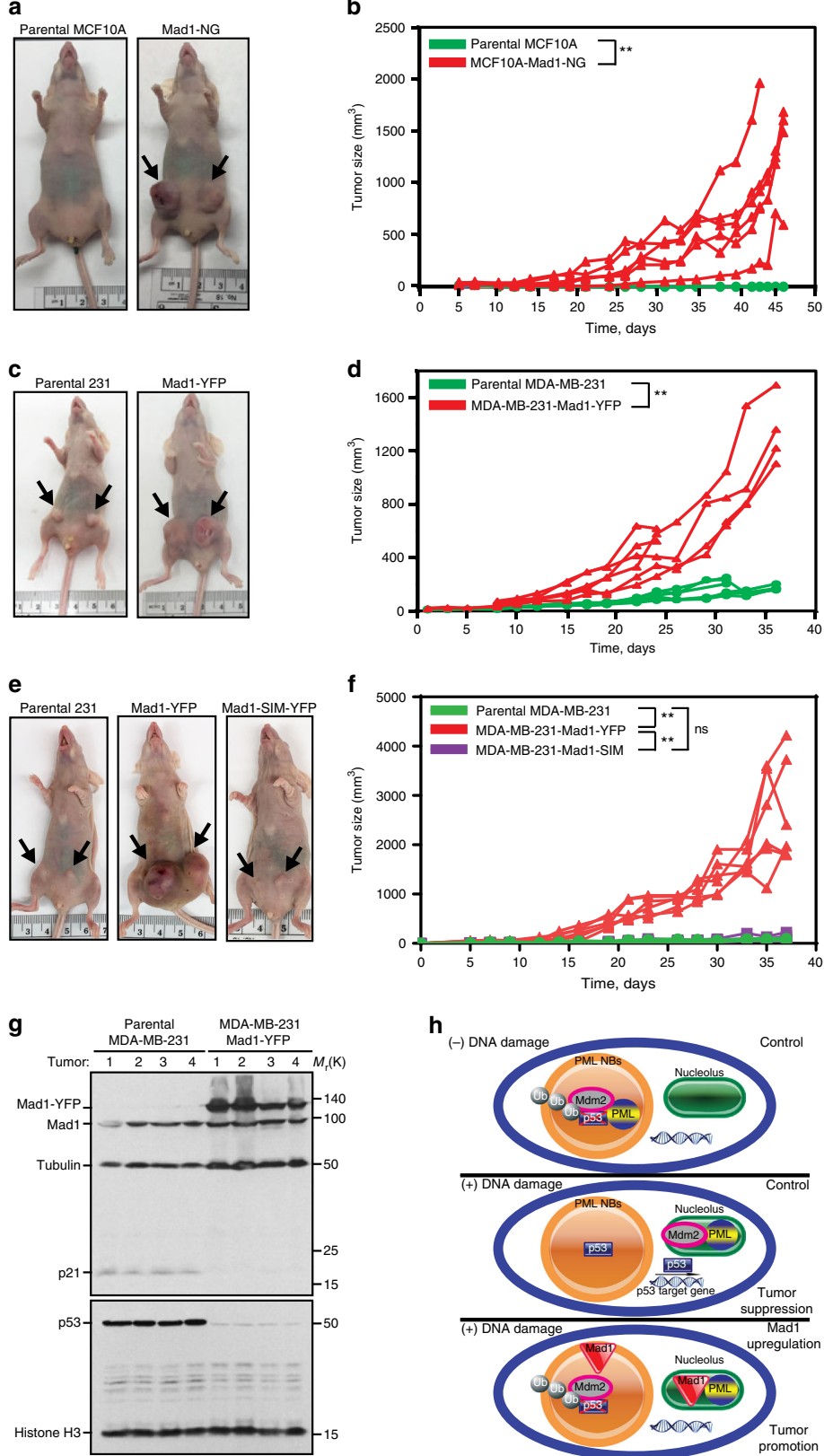

**Mad1 upregulation promotes tumors by inhibiting p53 and p21**. Mad1 is commonly upregulated at both the mRNA and protein levels in primary breast cancer[2]. To determine whether increased expression of Mad1 is sufficient to promote tumor initiation, nontransformed MCF10A cells were engineered to stably express Mad1-mNeonGreen in response to tet. Parental

cells that expressed the tet repressor but not Mad1-mNeonGreen were used as controls. Cells were injected orthotopically into mammary fat pads. All animals received chow containing the tet analog doxycycline (dox) to induce expression of Mad1-mNeonGreen or control for the effects of dox. Parental MCF10A cells formed tumors at 0 of 12 injection sites, while

**Fig. 7** Mad1 upregulation promotes tumor initiation and accelerates tumor growth. **a–f** Upregulation of Mad1 is tumor promoting. Parental cells express the tet repressor but not Mad1 in response to tet. All mice were on a diet containing the tet analog dox (and a blue dye) to induce expression of Mad1 or control for the effects of dox. Arrows indicate tumors. **a**, **b** Upregulation of Mad1 permits orthotopic mammary tumor development by MCF10A cells. **a** Representative images after orthotopic mammary gland injection of MCF10A cells into nude mice. Parental MCF10A cells are nontransformed and failed to form tumors at any of 12 injection sites in nude mice (control). However, expression of Mad1-mNeonGreen was sufficient to induce orthotopic tumor formation at 9 out of 12 injection sites. **b** Growth rates of tumors after injection of MCF10A cells. Data from 6 parental and 6 MCF10A-Mad1-NeonGreen contemporaneous injections from a single experiment are shown. **c**, **d** Expression of Mad1-YFP promotes the growth of MDA-MB-231 orthotopic mammary tumors in nude mice. **c** Representative images of orthotopic mammary tumors in nude mice. **d** Volumes of each MDA-MB-231 tumor over time. $n = 6$ parental and 6 Mad1-YFP expressing tumors. **e**, **f** PML binding via the C-terminal SIM is necessary for Mad1 to promote mammary tumor growth. **e** Representative images of orthotopic mammary tumors. **f** Volumes of each MDA-MB-231 tumor. $n = 6$ parental, 6 Mad-YFP and 6 SIM mutant Mad1 expressing tumors. **g** Tumors expressing Mad1-YFP have lower levels of p53 and its effector p21. Protein lysates from tumors collected at day 40 post-injection were analyzed by immunoblot. **h** A proposed model for the role of upregulated Mad1 in destabilizing p53. In unstressed control cells (top), p53 levels are low due to continuous ubiquitination by MDM2 in PML NBs. In control cells exposed to DNA damage (middle), PML sequesters MDM2 into the nucleolus, physically separating MDM2 from p53. p53 protein accumulates and induces transcription, resulting in tumor suppressive effects. When Mad1 is upregulated (bottom), Mad1 competes with MDM2 to bind PML. PML sequesters Mad1 into nucleoli in response to DNA damage, permitting MDM2 to continue to bind and ubiquitinate p53 within PML NBs. p53 protein levels remain low, and downstream events of p53 stabilization including p21 accumulation and cell death do not occur. $**p < 0.001$ by Sen–Adichie test. For specific $p$ values, see Source Data file

Mad1-mNeonGreen expressing MCF10A cells formed tumors at 9 of 12 injection sites (Fig. 7a, b). These data show that upregulation of Mad1 is sufficient for transformation and tumorigenesis in MCF10A cells.

To determine whether Mad1 upregulation promotes tumor growth and progression, MDA-MB-231 cells that express Mad1-YFP in response to tet were injected orthotopically into mammary fat pads. Parental MDA-MB-231 cells that express the tet repressor but not Mad1-YFP were used as controls. All animals received chow containing dox. As expected, parental MDA-MB-231 cells formed tumors at 6 of 6 injection sites. However, 6 of 6 MDA-MB-231 tumors expressing Mad1-YFP grew at a faster rate and to a larger size than all 6 of the parental tumors (Fig. 7c, d). In a separate experiment, MDA-MB-231 cells expressing SIM mutant Mad1 in response to tet grew at the same rate as parental cells, while all 6 tumors expressing Mad1-YFP substantially outgrew both the 6 parental and 6 SIM mutant Mad1-expressing tumors (Fig. 7e, f). Interestingly, while parental MDA-MB-231 tumors and tumors expressing SIM mutant Mad1 showed detectable levels of p53 and p21 expression, these were notably reduced in MDA-MB-231 tumors expressing Mad1-YFP (Fig. 7g, Supplementary Fig. 4d). These data demonstrate that upregulation of Mad1 destabilizes p53 and p21 in vivo and that p53 destabilization is central to the ability of Mad1 to promote orthotopic mammary tumor growth.

## Discussion
Mad1 plays a well-characterized role in regulating chromosome segregation during mitosis[2,3,57,6]. Interphase roles of Mad1 remain less well studied. Here we demonstrate that upregulated Mad1 prevents stabilization of p53 during interphase through direct binding to PML. p53 levels are normally kept low due to continuous ubiquitination by MDM2 in PML NBs (Fig. 7h, top)[28,30,58]. In response to DNA damage, PML sequesters MDM2 into nucleoli (Fig. 7h, middle)[31,33,34,56]. This spatially separates MDM2 from p53, stabilizing p53 protein and allowing it to regulate gene transcription, resulting in p21 expression, cell cycle arrest, and/or cell death. Our data support a model in which upregulated Mad1 directly binds PML through its CTD in a manner promoted by sumoylation. Mad1 binding displaces MDM2 from PML. PML then sequesters Mad1 into nucleoli (Fig. 7h, bottom). Displaced MDM2 remains in close physical proximity to p53, allowing it to bind and ubiquitinate p53, leading to continuous p53 degradation and tumor promotion. Importantly, this mechanism regulates the expression of both

wild type and mutant p53. These data provide molecular insight into an unexpected interphase role of Mad1.

Mad1 is commonly upregulated in human breast cancer at both the mRNA and protein levels, where it serves as a marker of poor prognosis[2,59]. Mad1 upregulation enhances anchorage-independent growth in culture[2] and promotes both tumor initiation and tumor growth in orthotopic models of breast cancer (Fig. 7a–f). Upregulation of Mad1 to levels similar to those found in human breast cancer causes a low rate of chromosome missegregation, which is weakly tumor promoting[2,22–24,60]. However, p53 is the most commonly altered tumor suppressor in human cancers, and destabilization of p53 substantially contributes to the tumor-promoting effects of Mad1 upregulation (Fig. 7e, f).

It is interesting that Mad1 expression regulates p53 levels not only in cells with a wild type p53 pathway (MCF10A), but also in cells in which p53 function is altered due to mutation (S241F in DLD1; R280K in MDA-MB-231) or by human papillomavirus E6 (HeLa). Our results are in agreement with previous studies showing that even mutant p53 is stabilized in response to DNA damage[61,62]. This suggests that Mad1 destabilization of p53 is robust and widespread, and is likely to occur in most tumor cells in which Mad1 is upregulated, independent of p53 mutation status.

Numerous mouse models with altered expression of mitotic checkpoint genes have been used as presumptive tests of the impact of aneuploidy and chromosomal instability (CIN) on tumor initiation[14,63–65]. Taken together, these studies support the overarching conclusion that low rates of CIN are weakly tumor promoting while high rates of CIN lead to cell death and tumor suppression[22–24,60]. However, the tumor phenotypes of mouse models with distinct genetic manipulations vary widely despite similar levels of aneuploidy. Like Mad1, most mitotic checkpoint genes are expressed throughout the cell cycle and have been implicated in diverse cellular functions during interphase[22,66]. The previously unsuspected role of Mad1 in influencing p53 stability has a substantial impact on the orthotopic tumor phenotype caused by Mad1 upregulation. Thus, the wide variety of tumor phenotypes caused by alteration of "mitotic" genes is likely to be due in large part to the known as well as the undiscovered interphase functions of these genes.

The importance of sumoylation in chromosome segregation during mitosis is emerging, with roles for sumoylation identified on several critical mitotic regulators, including CENP-E, BubR1, and APC/C[46,47,67–69]. Here we demonstrate a functional role for the Mad1 SIM during interphase. Whether sumoylation or

SUMO interaction affects the interphase function of additional "mitotic" genes as well as whether sumoylation and binding affect the mitotic function of Mad1 remain important unanswered questions.

## Methods

**Antibodies**. Antibodies used in this study include affinity-purified rabbit anti-Mad1 antibodies prepared against amino acids 333–617 of human Mad1[2] and diluted 1:3000 for western blots and 1:2000 for immunofluorescence, mouse anti-SUMO1 (21C7, Developmental Studies Hybridoma Bank) diluted 1:250, mouse anti-Myc (9E10, Developmental Studies Hybridoma Bank) diluted 1:200, mouse anti-tubulin (12G10, Developmental Studies Hybridoma Bank) diluted 1:5000, rabbit anti-GFP (#2555S, Cell Signaling), mouse anti-p53 (DO-1, #sc-126, Santa Cruz), rabbit anti-p53 (#NB200-171, Novusbio), mouse anti-actin (JLA20, Developmental Studies Hybridoma Bank) diluted 1:500, rabbit anti-tubulin (#2144S, Cell Signaling), rabbit anti-p21 (#ab188224, Abcam), rabbit anti histone H3 (# 9715S, Cell Signaling), mouse anti-FLAG M2 (Sigma) diluted 1:2500, rabbit anti-PML (sc-5621) diluted 1:500, and mouse anti-HA (#901501, Biolegend), and pan-cytokeratin antibody (AE1 + AE3 conjugated to Alexa Fluor® 647, Novus, cat. no. NBP2-33200AF647).

**Cell culture**. HeLa (ATCC, CCL-2) and DLD1 (ATCC CCL-221) cells were grown in DMEM with 10% FBS, 100 U/mL penicillin and 100 mg/mL streptomycin and cultured in 5% $CO_2$ at 37 °C. MDA-MB-231 (ATCC HTB-26) cells were grown in DMEM with 10% FBS supplemented with 100 U/mL penicillin and 100 mg/mL streptomycin and cultured in 10% $CO_2$ at 37 °C. MCF10A cells (ATCC CRL-10317) were grown in DMEM/F12 with 5% horse serum, 100 U/mL penicillin, 100 mg/mL streptomycin, 20 ng/mL EGF, 0.5 mg/mL hydrocortisone, 100 ng/mL cholera toxin and 10 μg/mL insulin and cultured in 5% $CO_2$ at 37 °C. SKBR3 (ATCC HTB-3) cells were grown in ATCC-formulated McCoy's 5a Medium Modified with 10% FBS supplemented with 100 U/mL penicillin and 100 mg/mL streptomycin and cultured in 5% $CO_2$ at 37 °C. Stable cell lines were generated by transducing cells stably expressing the tet repressor with retroviruses expressing full length wild type Mad1-3xFLAG, Mad1(ΔCTD)-3xFLAG, or Mad1(SIM mutant)-3xFLAG under a tet-inducible promoter. Stable integrants were selected with 4 μg/mL puromycin and 200 μg/mL blasticidin and validated for inducible expression of Mad1 upon tet addition. Virus expressing shRNA against Mad1 (5′-TGAGATCTTTGAACAACTT-3′ for Mad1-shRNA#40, 5′-AGCGATTGTGA AGAACATG-3′ for Mad1-shRNA#717, 5′-GCTTGCCTTGAAGGACAAG-3′ for shRNA#1036, 5′-GCGATTGTGAAGAACATGA-3′ for ad1- shRNA#718) was made from pSUPERIOR.retro.puro. SKBR3 cells were infected by virus containing shRNA sequence against Mad1 and stable integrants were selected with 4 μg/mL puromycin.

**Plasmids, mutagenesis, cloning, and virus production**. Cloning was performed using Gibson assembly. The plasmids and primers used in this study are listed in Supplementary Table 1. Full length and truncated Mad1 with 3xFLAG or HA tag were cloned into the pRetro-CMV2-TO-puromycin vector. The SIM mutation of Mad1 was generated by site-directed mutagenesis. The pCMV-HA-PML-IV plasmid was a kind gift from Drs. Moon Hee Lee and Shigeki Miyamoto. Nucleolin-GFP and 3xFLAG-MDM2 cDNAs linked by P2A were cloned into pRetro-CMV2-TO-Hygromycin vector. Retroviruses were first generated by transient transfection of HEK 293T cells with the pReto-CMV-TO and separate plasmids that express Gag-Pol, Rev, Tat, and VSV-G. Supernatants were clarified by filtration. Nucleolin-mNeonGreen and mScarlet-MDM2 cDNAs linked by P2A, Mad1-mNeonGreen, and mNeonGreen were cloned into pENTR1A vector (Invitrogen), which contains a CMV promoter, and SV40 polyA signal, and introduced into the pAd/PL-DEST vector by using LR Clonase (Invitrogen). HEK293A cells were transfected with pAd/PL-DEST vectors after linearization with PacI. Viral particles were isolated by three freeze–thaw cycles and amplified by reinfection of HEK293A cells.

**Immunostaining of tumor tissues**. De-identified formalin-fixed paraffin-embedded human breast cancer tissue sections were obtained from the Translational Research Initiatives in Pathology Lab at UW-Madison (IRB exempt). Samples were deparaffinized and rehydrated by sequential incubation in xylene (3 × 5 min), 100% ethanol (3 × 5 min), 95% ethanol (3 × 5 min), 75% ethanol (2 × 5 min), and $H_2O$ (2 × 5 min). Antigen retrieval was performed by heating for 30 min at 95 °C in 10 mM citrate buffer (pH 6.0). Samples were cooled for at least 30 min and washed 3 × 5 min with 1× TBS/0.3% Triton X-100. Tissues were delineated using a hydrophobic barrier pen. Tissues were blocked for 24 h at 4 °C with 1× TBS/5% goat serum and 5% BSA, then incubated overnight at 4 °C with primary antibodies diluted in 5% goat serum in 1× TBS/0.3% Triton X-100. Antibodies used were anti-Mad1 rabbit polyclonal (1:2000) and anti-SUMO1 mouse monoclonal (1:250). Tissues were washed 3 × 10 min in 1× TBS/0.3% Triton X-100 followed by incubation with FITC and TRITC labeled secondary antibodies for 1 h. Tissues were then washed 5 × 10 min in 1× TBS/0.3% Triton X-100 and incubated overnight at 4 °C with pan-cytokeratin antibody. Tissues were washed 3 × 10 min with 1× TBS/0.3% Triton X-100 before mounting with DAPI.

**Immunoprecipitation**. $5 × 10^6$ HEK-293T cells were seeded in 10 cm cell culture dishes and grown for 24 h. Cells were transiently transfected using calcium phosphate with 25 μg plasmid. Cells were scraped in RIPA buffer (50 mM Tris–HCl, pH 8.0, 1 mM EDTA, 150 mM NaCl, 20% glycerol, 1% NP-40, 0.5% sodium deoxycholate, and 1× PMSF) and cell lysates kept on ice for 30 min. Cell lysates were cleared by centrifugation at $10,000 × g$ for 1 min and immunoprecipitations were performed with anti-FLAG M2 affinity gel (No. A2220-5ML, Sigma-Aldrich) or anti-HA magnetic beads (No. B26202, Bimake) for 5 h at 4 °C. Samples were washed 5 × 30 min in RIPA buffer. Proteins of immunoprecipitates and total cell lysates were separated by 12% SDS-PAGE, transferred to nitrocellulose membranes, blocked with 5% milk in TBST and analyzed by immunoblotting with indicated antibodies.

**Immunofluorescence and immunoblotting**. Cells were rinsed in PBS, fixed for 10 min in 4% formaldehyde, and permeabilized for 7 min in 0.25% Triton X-100 in PBS. Unless otherwise specified, images were acquired on a Nikon Ti-E inverted microscope using a CoolSNAP HQ2 camera driven by Nikon Elements software and subsequently deconvolved using the AQI 3D Deconvolution module in Nikon Elements. 2D maximum projections assembled in Elements are shown. Overlays were generated in Photoshop. For immunoblotting, equal numbers of cells were lysed in 2× sample buffer. Proteins were separated by 12% SDS-PAGE, transferred to nitrocellulose, blocked with 5% milk in TBST and probed with primary and secondary antibodies.

**MBP pull-down experiments**. Protein expression was carried out in BL21 *Escherichia coli* cotransformed with pMAL-c2x (Mad-1, Mad-1 CTD, MBP alone) and pET-28a (Smt3p-PML, Smt3p-3xSUMO2-PML). The cells were sonicated in lysis buffer (50 mM HEPES pH 7.6, 100 mM NaCl, 1 mM DTT) and the lysate obtained after centrifugation was incubated with SUMO protease overnight at 4 °C to remove the Smt3p tag. The lysate was then incubated with amylose resin for 1 h at 4 °C. After washing with lysis buffer, bound proteins were eluted from the resin with elution buffer (50 mM HEPES pH 7.6, 100 mM NaCl, 1 mM DTT, 10 mM maltose) and analyzed via immunoblot.

**Nuclear and cytoplasmic protein fractionation**. Cells from a 10 cm dish were harvested and incubated in Buffer A (200 mM HEPES pH 7.9, 400 mM NaCl, 1 mM EDTA, 1 mM EGTA, pH 7.4) with protease inhibitors for 10–15 min on ice. NP-40 was added to a final concentration of 0.5%. Lysate was centrifuged at $2000 × g$ for 5 min to separate supernatant (cytoplasm) and pellet (nuclei).

**Orthotopic mouse model**. All animal studies were performed in compliance with all relevant ethical regulations for animal testing and research. The study was approved by the Institutional Animal Care and Use Committee of the University of Wisconsin-Madison. $2.5 × 10^6$ MDA-MB-231 or $5 × 10^6$ MCF10A cells were injected into mammary fat pads of 5-week-old female athymic nude mice. Mice were fed Teklad TD.120769, which contains 625 mg/kg doxycycline. Tumor size was measured every 2 days using calipers. Tumor volumes were calculated using the formula $v = width^2 × length/2$. After 7 weeks, necropsy was performed, and tumors were harvested for protein isolation.

**Statistical analysis**. The Sen–Adichie test calculated in MTSAT (https://mcardle.wisc.edu/mstat/), version 6.4.2 was used to compare tumor volumes. For all other experiments, significant differences were determined using a two-tailed Student $t$ test calculated using the R commander (Rcmdr) package in R (https://socialsciences.mcmaster.ca/jfox/Misc/Rcmdr/) version 2.4-4. Results are presented as mean ± SD unless otherwise specified.

**Reporting summary**. Further information on experimental design is available in the Nature Research Reporting Summary linked to this article.

## Data availability

The authors declare that all data supporting the findings of this study are available within the paper and its Supplementary Information files. The source data underlying Figs. 1f, g, i, 2b, c, h, i, 3d, f, 4a–e, 5b, d, f, 6a–c, 7b, d f, g and Supplementary Figs. 1b, 1e, 1i-m, 2a-b, 3b, 4a-d and 5b-c are provided as a Source Data file.

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

## Acknowledgements

The authors thank Dr. Moon Hee Lee and Dr. Shigeki Miyamoto for the PML-IV cDNA, the Biomedical Research Model Services Mouse Breeding Core for use of its facilities and services, and members of the Burkard and Weaver laboratories for useful discussions. The authors also thank the University of Wisconsin Translational Research Initiatives in Pathology laboratory, in part supported by the UW Department of Pathology and Laboratory Medicine and UWCCC Grant P30 CA014520, for use of its facilities and services. Beth A. Weaver was supported by a Research Scholar Grant, RSG-15-006-01 - CCG, from the American Cancer Society. This work was also supported, in part, by predoctoral fellowship 16PRE29650011 from the American Heart Association (J.W.) and NIH Grants GM088151 (A.A.) and T32CA009135 (C.S.).

## Author contributions

J.W. and B.A.W. conceived the project. J.W., S.B., C.M.S., and R.T. performed experiments and collected data. K.E. and A.A. provided critical expertise and resources. J.W., S.B., and B.A.W. wrote the manuscript. All authors provided feedback on the manuscript before submission.

## Additional information

**Competing interests:** The authors declare no competing interests.

