## [Peer Review File · Nature Communications]

Reviewers' comments:

Reviewer #1 (Remarks to the Author):

The paper addresses an open question in the field, unveiling the role of Mad1 protein in interphase, as opposed to the well characterized role in chromosome segregation. In detail, they describe a model –appropriately supported by experimental data- in which upregulated Mad1, as commonly occurs in human breast cancer, interferes with p53 signalling, by binding PML through its CTD (binding enhanced by sumoylation) displacing MDM2 from PML, which in turn sequesters Mad1 into nucleoli. MDM2 is thus "free" to bind p53 and initiate its degradation, ultimately promoting tumorigenesis.

The subject of the paper is novel and of interest to the field, as it demonstrates the existence a novel mechanism of regulation/de-regulation of the p53 pathway. Being p53 the most commonly altered tumor suppressor in human cancer, having a comprehensive understanding of the related molecular mechanisms is indeed critical.

Though the finding is unlikely to represent a breakthrough in the field and radically change the thinking in the field, it adds an important piece of information, supported by well-designed, convincing experimental data; a large body of results, indeed, fully support the assertions made by the authors. Methods section is exhaustive, providing all the required information to potentially reproduce the experiments, even though additional details on the percentage of SDS-PAGE gel used for the western blot might be useful, as well as the composition of the blocking solution (as these may affect antibody efficiency).

Minor points:

In figure legend of figure 1, scale bar of panel D is missing. Panel G-H, the rationale for the experiments is provided in the main text, thus unnecessary in the figure legend. It can be removed.

Figure 2 panel B. Authors claim only fragments of Mad1 containing CTD domain (aa 597-718, and so also all the fragments containing the 597-718 aminoacidic stretch) co immunoprecipitate HA PML, but a signal seems slightly visible also in the lane co immunoprecipitating the fragment 1-178. It would be nice to see another blot.

Figure 2 panel F, having the acronym (NIS, NES, CTD ...) on the figure rather than the aminoacid sequence might be helpful for a more straightforward look of the data.

Figure 4 panel A, is the labelling of the second half of the blot correct ((+) (-) tet) ?

Overall, some details about the procedure should be moved from the figure legend to the methods or results section where it is more appropriate (see also figure 4E).

Reviewer #2 (Remarks to the Author):

In this manuscript Wan et al investigate the consequences of overexpressing Mad1, a known spindle checkpoint protein. The authors report here that when overexpressed Mad1 has not only a function in mitosis, but also changes the behavior of interphasic cells. This is potentially relevant as Mad1 has been often reported to be overexpressed in cancer cells. Specifically the authors show with biochemical and cell biology experiments that overexpressed Mad1 accumulates in PML bodies via Sumoylation interaction motif in the C-terminal part of the protein. This interaction displaces Mdm2 from PML bodies preventing the accumulation of p53 in the presence of DNA damage. In the last part of the manuscript the authors show with murine models that Mad1 overexpression can transform non-cancerous MCF10A cells and accelerate cancer formation when overexpressed in cancerous cells expressing a mutated version of p53. Based on these experiments they infer that the Mad1-overexpression seen in human cancer patients can participate to cancer formation.

Overall, this study addresses an interesting and original biological question. The results are novel and of high technical quality, in particular the protein-protein interaction experiments. The study has the potential for an excellent publication in Nature Communication, yet at this stage suffers

from several conceptual weaknesses that should be addressed before publication.

Major points:

1) the entire study is based on the strong overexpression of Mad1 in different cell lines. It is however, unclear whether Mad1 is really overexpressed to such high levels in cancer cells, raising the question whether the observed effects ever occur in a human cancer. The most convincing way to address this point would be for the patient to identify a tissue culture cell line that overexpresses Mad1 into PML bodies, and to quantify the p53 response after control or Mad1 depletion. Such an experiment would go a long way to prove that the Mad1 overexpression seen in cancer patients does participate to p53-regulation, and that the observed results are physiologically relevant.

2) the other important aspect is that the authors assume that Mad1 overexpression participates to cancerogenesis because it targets p53. This is plausible based on their data, but not directly proven in the manuscript. In fact Mad1 overexpression also has an effect in a cell line (MDA-MB-231) in which the p53 response is seriously attenuated, since it expresses a potentially dominant-negative p53 mutant. An important control would have been to also overexpress Mad1 in a cancer cell line bearing a p53 deletion, to show that in such case Mad1 overexpression has no effect on cancerogenesis, as p53 is missing. Since such an experiment would take a long time, I don't think it is necessary for the revision of this manuscript. Nonetheless if the authors had data that would address this point, this would significantly strengthen the conclusions. Otherwise the authors should at least comment on this question in their discussion.

Minor point:

1) since MDA-MB-231 cells express a mutant version of p53 that is strongly attenuated in its function, one would expect high levels of p53, due to low p53-dependent transcription of the MDM2 gene, which is what has been reported in the past. It is therefore surprising that the authors see a similar p53 accumulation in MDA-MB-231 and non-cancerous MCF10A cells after DNA damage, since p53 is supposed to be only very partially active and expressed at high levels in MDA-MB-231 cells.

Patrick Meraldi

Point-by-point response to the referees' comments

Reviewer #1 (Remarks to the Author):

The paper addresses an open question in the field, unveiling the role of Mad1 protein in interphase, as opposed to the well characterized role in chromosome segregation. In detail, they describe a model –appropriately supported by experimental data- in which upregulated Mad1, as commonly occurs in human breast cancer, interferes with p53 signalling, by binding PML through its CTD (binding enhanced by sumoylation) displacing MDM2 from PML, which in turn sequesters Mad1 into nucleoli. MDM2 is thus "free" to bind p53 and initiate its degradation, ultimately promoting tumorigenesis.

The subject of the paper is novel and of interest to the field, as it demonstrates the existence a novel mechanism of regulation/de-regulation of the p53 pathway. Being p53 the most commonly altered tumor suppressor in human cancer, having a comprehensive understanding of the related molecular mechanisms is indeed critical.

Though the finding is unlikely to represent a breakthrough in the field and radically change the thinking in the field, it adds an important piece of information, supported by well-designed, convincing experimental data; a large body of results, indeed, fully support the assertions made by the authors. Methods section is exhaustive, providing all the required information to potentially reproduce the experiments, even though additional details on the percentage of SDS-PAGE gel used for the western blot might be useful, as well as the composition of the blocking solution (as these may affect antibody efficiency).

We thank the Reviewer for recognizing the importance, novelty, and experimental rigor of our manuscript as well as for making useful suggestions to improve it. Additional details about gel percentages and composition of the blocking solution have been added to the methods section.

Minor points:

In figure legend of figure 1, scale bar of panel D is missing.

Thank you for bringing this to our attention. It has been added.

Panel G-H, the rationale for the experiments is provided in the main text, thus unnecessary in the figure legend. It can be removed.

This portion of the figure legend was removed.

Figure 2 panel B. Authors claim only fragments of Mad1 containing CTD domain (aa 597-718, and so also all the fragments containing the 597-718 aminoacidic stretch) co immunoprecipitate HA PML, but a signal seems slightly visible also in the lane co immunoprecipitating the fragment 1-178. It would be nice to see another blot.

The reason the band is visible is because 1-718 represents the full length protein. We have now designated this lane as 1-718 (FL) to make this clear.

Figure 2 panel F, having the acronym (NIS, NES, CTD ...) on the figure rather than the aminoacid sequence might be helpful for a more straightforward look of the data.

These changes have been made.

Figure 4 panel A, is the labelling of the second half of the blot correct ((+) (-) tet) ?

Indeed, it is not. Thank you (again) for catching this. It has been fixed.

Overall, some details about the procedure should be moved from the figure legend to the methods or results section where it is more appropriate (see also figure 4E).

The figure legends have been edited to remove redundancy and be more succinct.

Reviewer #2 (Remarks to the Author):

In this manuscript Wan et al investigate the consequences of overexpressing Mad1, a known spindle checkpoint protein. The authors report here that when overexpressed Mad1 has not only a function in mitosis, but also changes the behavior of interphasic cells. This is potentially relevant as Mad1 has been often reported to be overexpressed in cancer cells. Specifically the authors show with biochemical and cell biology experiments that overexpressed Mad1 accumulates in PML bodies via Sumoylation interaction motif in the C-terminal part of the protein. This interaction displaces Mdm2 from PML bodies preventing the accumulation of p53 in the presence of DNA damage. In the last part of the manuscript the authors show with murine models that Mad1 overexpression can transform non-cancerous MCF10A cells and accelerate cancer formation when overexpressed in cancerous cells expressing a mutated version of p53. Based on these experiments they infer that the Mad1-overexpression seen in human cancer patients can participate to cancer formation.

Overall, this study addresses an interesting and original biological question. The results are novel and of high technical quality, in particular the protein-protein interaction experiments. The study has the potential for an excellent publication in Nature Communication, yet at this stage suffers from several conceptual weaknesses that should be addressed before publication.

We thank the Reviewer for noting the importance, novelty and high technical quality of our work. We are also grateful for pointing out the weaknesses of the study, which we have addressed in the revised version. We believe the additional experiments improve the manuscript and will increase its impact.

Major points:

1) the entire study is based on the strong overexpression of Mad1 in different cell lines. It is however, unclear whether Mad1 is really overexpressed to such high levels in cancer cells, raising the question whether the observed effects ever occur in a human cancer. The most convincing way to address this point would be for the patient to identify a tissue culture cell line that overexpresses Mad1 into PML bodies, and to quantify the p53 response after control or Mad1 depletion. Such an experiment would go a long way to prove that the Mad1 overexpression seen in cancer patients does participate to p53-regulation, and that the observed results are physiologically relevant.

As the Reviewer suggested, we identified that Mad1 is 'naturally' overexpressed in SKBR3 breast cancer cells (new figure S5a). As expected, Mad1 colocalizes with PML nuclear bodies in this cell line (new figure S5b). Partial depletion of Mad1 using four distinct shRNA sequences increased expression of p53 after DNA damage induced by doxorubicin (new figure S5c). These results are precisely what our model predicted and are discussed on pages 8-9 of the revised manuscript. We agree with the Reviewer that they provide strong support for the physiological relevance of our findings.

2) the other important aspect is that the authors assume that Mad1 overexpression participates to cancerogenesis because it targets p53. This is plausible based on their data, but not directly proven in the manuscript. In fact Mad1 overexpression also has an effect in a cell line (MDA-MB-231) in which the p53 response is seriously attenuated, since it expresses a potentially dominant-negative p53 mutant. An important control would have been to also overexpress Mad1 in a cancer cell line bearing a p53 deletion, to show that in such case Mad1 overexpression has no effect on cancerogenesis, as p53 is missing. Since such an experiment would take a long time, I don't think it is necessary for the revision of this manuscript. Nonetheless if the authors had data that would address this point, this would significantly strengthen the conclusions. Otherwise the authors should at least comment on this question in their discussion.

The Reviewer is correct that these experiments require a substantial amount of time. Fortunately, we had begun an experiment similar to the one the Reviewer suggests, which we believe speaks to this point. In this experiment, we showed that MDA-MB-231 cells that express SIM mutant Mad1 – which does not bind PML or destabilize p53 – form orthotopic tumors that grow at similar rates as parental MDA-MB-231 cells. These data provide strong support for the importance of p53 destabilization in the tumor promoting activity of Mad1 upregulation. They are now shown in Figure 7e-f and discussed on page 11 of the revised manuscript.

Minor point:

1) since MDA-MB-231 cells express a mutant version of p53 that is strongly attenuated in its function, one would expect high levels of p53, due to low p53-dependent transcription of the MDM2 gene, which is what has been reported in the past. It is therefore surprising that the authors see a similar p53 accumulation in MDA-MB-231 and non-cancerous MCF10A cells after DNA damage, since p53 is supposed to be only very partially active and expressed at high levels in MDA-MB-231 cells.

Though MDA-MB-231 cells express an R280K mutant of p53, our data are in agreement with other laboratories, which have previously shown that DNA damage caused by doxorubicin is sufficient to increase p53 protein levels in these cells (Carroll et al, Oncotarget 2016; Xu and Loo, J Cell Biochem. 2001). This suggests that even a low level of MDM2 is capable of partially reducing the level of mutant p53. It is indeed interesting that Mad1 regulation of p53 occurs not only in cells with a wild type p53 pathway but also in cells expressing mutant p53 or HPV E6. This is now discussed on pages 12-13 of the revised manuscript.

REVIEWERS' COMMENTS:

Reviewer #1 (Remarks to the Author):

The paper addresses an open question in the field, unveiling the role of Mad1 protein in interphase, as opposed to the well characterized role in chromosome segregation during mitosis. In detail, they describe a model –appropriately supported by experimental data- in which upregulated Mad1, as commonly occurs in human breast cancer, interferes with p53 pathway, by binding PML through its CTD (binding enhanced by sumoylation) displacing MDM2 from PML, which in turn sequesters Mad1 into nucleoli. MDM2 is thus "free" to bind p53 and initiate its degradation, ultimately promoting tumorigenesis.

The subject of the paper is novel and of interest in the field, as it demonstrates the existence and the role of a protein in controlling the p53 pathway. Being p53 the most commonly altered tumor suppressor in human cancer, having a comprehensive understanding of the related molecular mechanisms is important.

Though the finding is unlikely to represent a breakthrough in the field and radically change the thinking in the field, it adds an important piece of information, supported by well-designed, convincing experimental data; a large body of results, indeed, fully support the assertions made by the authors. Methods section is exhaustive, providing all the required information to potentially reproduce the experiments, even though additional details on the percentage of SDS-PAGE gel used for the western blot might be useful, as well as the composition of the blocking solution (as these may affect antibody efficiency).

Minor points:

In figure legend 1, scale bar of panel D is missing. Panel G-H, the rationale for the experiments is provided in the main text, thus unnecessary in the figure legend. It can be removed.

Figure 2 panel B. Authors claim only fragments of Mad1 containing CTD domain (aa 597-718, and so also all the fragments containing the 597-718 aminoacidic stretch) co immunoprecipitate HA PML, but a signal seems slightly visible also in the lane co immunoprecipitating the fragment 1-178. It would be nice to see another blot.

Figure 2 panel F, having the acronym (NIS, NES, CTD ...) on the figure rather than the aminoacid sequence might be helpful for a more straightforward look of the data.

Figure 4 panel A, the second half of the blot should be (+) tet rather than (-) tet

Reviewer #2 (Remarks to the Author):

The authors have addressed the reviewers concerns in a convincing manner. I therefore support publication of this interesting study.

Patrick Meraldi

REVIEWERS' COMMENTS:

Reviewer #1 (Remarks to the Author):

The paper addresses an open question in the field, unveiling the role of Mad1 protein in interphase, as opposed to the well characterized role in chromosome segregation during mitosis. In detail, they describe a model –appropriately supported by experimental data- in which upregulated Mad1, as commonly occurs in human breast cancer, interferes with p53 pathway, by binding PML through its CTD (binding enhanced by sumoylation) displacing MDM2 from PML, which in turn sequesters Mad1 into nucleoli. MDM2 is thus "free" to bind p53 and initiate its degradation, ultimately promoting tumorigenesis.

The subject of the paper is novel and of interest in the field, as it demonstrates the existence and the role of a protein in controlling the p53 pathway. Being p53 the most commonly altered tumor suppressor in human cancer, having a comprehensive understanding of the related molecular mechanisms is important.

Though the finding is unlikely to represent a breakthrough in the field and radically change the thinking in the field, it adds an important piece of information, supported by well-designed, convincing experimental data; a large body of results, indeed, fully support the assertions made by the authors. Methods section is exhaustive, providing all the required information to potentially reproduce the experiments, even though additional details on the percentage of SDS-PAGE gel used for the western blot might be useful, as well as the composition of the blocking solution (as these may affect antibody efficiency).

We thank the Reviewer for recognizing the importance, novelty, and experimental rigor of our manuscript as well as for making useful suggestions to improve it. Additional details about gel percentages and composition of the blocking solution have been added to the methods section.

Minor points:

In figure legend 1, scale bar of panel D is missing.

Thank you for bringing this to our attention. It has been added.

Panel G-H, the rationale for the experiments is provided in the main text, thus unnecessary in the figure legend. It can be removed.

This portion of the figure legend has been removed.

Figure 2 panel B. Authors claim only fragments of Mad1 containing CTD domain (aa 597-718, and so also all the fragments containing the 597-718 aminoacidic stretch) co immunoprecipitate HA PML, but a signal seems slightly visible also in the lane co immunoprecipitating the fragment 1-178. It would be nice to see another blot.

The reason the band is visible is because 1-718 represents the full length protein. We have now designated this lane as 1-718 (FL) to make this clear.

Figure 2 panel F, having the acronym (NIS, NES, CTD ...) on the figure rather than the aminoacid

sequence might be helpful for a more straightforward look of the data.

These changes have been made.

Figure 4 panel A, the second half of the blot should be (+) tet rather than (-) tet

Thank you for catching this. It has been fixed.

Reviewer #2 (Remarks to the Author):

The authors have addressed the reviewers concerns in a convincing manner. I therefore support publication of this interesting study.

We thank the Reviewer for this review and for previous suggestions to improve the manuscript.